# 2D Short-Time Fourier Transform for local morphological analysis of meibomian gland images

Kamila Ciężar[1,2], Mikolaj Pochylski[1]*

1 Faculty of Physics, Adam Mickiewicz University, Poznań, Poland, 2 Augenarztpraxen Berlin Suedwest, Berlin, Germany

* pochyl@amu.edu.pl

## Abstract

Meibography is becoming an integral part of dry eye diagnosis. Being objective and repeatable this imaging technique is used to guide treatment decisions and determine the disease status. Especially desirable is the possibility of automatic (or semi-automatic) analysis of a meibomian image for quantification of a particular gland's feature. Recent reports suggest that in addition to the measure of gland atrophy (quantified by the well-established "drop-out area" parameter), the gland's morphological changes may carry equally clinically useful information. Here we demonstrate the novel image analysis method providing detailed information on local deformation of meibomian gland pattern. The developed approach extracts from every Meibomian image a set of six morphometric color-coded maps, each visualizing spatial behavior of different morphometric parameter. A more detailed analysis of those maps was used to perform automatic classification of Meibomian glands images. The method for isolating individual morphometric components from the original meibomian image can be helpful in the diagnostic process. It may help clinicians to see in which part of the eyelid the disturbance is taking place and also to quantify it with a numerical value providing essential insight into Meibomian gland dysfunction pathophysiology.

## Introduction

Assessment of Meibomian glands (MGs) condition has been the focus of many studies in recent years [1, 2]. This interest results from the fact that dysfunction in MG physiology is a leading factor of the dry eye disease with the prevalence that varies widely from 3,5% to 70% based on the age, sex and ethnicity [3, 4]. The most common diagnosis of MGD is based on subjective symptoms and more detailed examination of the anterior eye structures [5–7]. These early subjective methods of the MG state classification (basing on the personal experience of the specialist and characterized by high inconsistent and low repeatability) are slowly being replaced with the sophisticated semiautomatic and automatic image analyzing methods [8–18]. Providing standard quantification of the gland structure, these methods open the possibility for increased measurement repeatability and shortening the diagnostic time [19–29]. Not surprisingly, according to the latest reports [10–16] there is a strong need to develop new image analysis protocols.

**Data Availability Statement:** All relevant data are within the paper and its Supporting information files.

**Funding:** The author(s) received no specific funding for this work.

**Competing interests:** The authors have declared that no competing interests exist.

A well-known objective measure of MGs condition is the "drop-out area" (DOA) which quantifies the meibomian gland loss [30]. It is defined as the ratio between the area covered by meibomian glands and the total eyelid area. This simple definition makes the value of DOA relatively easy to estimate automatically, directly from the meibomian image [19, 20]. The easy and intuitively understandable definition of DOA makes this parameter the most frequently used objective measure of MGs condition utilized in quantification of MGD progression. The other and less obvious MGD symptoms which relate to more subtle changes in the gland morphology with no visible changes in meibomian gland loss. Recently, it has been shown that apart from the gland atrophy, the distortion in gland's shape is a valuable complementary clinical feature of MGs [12]. Although the mechanism of the MGD progression is still unclear and the data regarding gland tortuosity in the general population is still needed [9], a clear correlation between MGs deformation and clinical parameters such as meibum expressibility, lid margin score, meiboscore, meibum expressibility score, and TBUT has been demonstrated [9, 12, 31]. It is thus believed that the MGD progresses from an early stage characterized by subtle gland distortion, whereas the loss of MGs is observed only in the advanced stage of the disease [9, 12, 31, 32]. In this perspective the MG's shape distortion may be considered as an early indicator of various ophthalmic diseases (including MGD and dry eye syndrome) which is a strong motivation for development of methods for objective examination and parametrization of MGs deformation. The ability to parametrize the local morphology of MGs should allow to use these measures (in addition to DOA) for better assistance in the diagnosis process and MGD severity evaluation.

Unfortunately, the objective description of the gland deformity is much more difficult than just determining the degree of its atrophy (measured by DOA). There are several works introducing the meibomian gland classification based on their deformation, but so far the gold standard has not been established [9–12, 24–27]. In searching for other objective descriptors of MG morphological condition, we have recently presented an approach for quantifying and classifying Meibomian images using 2D Fourier Transform (2DFT) [33]. This global analysis, performed on the whole set of the glands, demonstrated that information on mean gland frequency (connected with mean width of glands or inter-gland section) and anisotropy in gland periodicity (related to mean spread in gland directions) can be used for automatic image classification. However, despite currently being global, the method can be blind to some important slight local disturbances in gland patterns. Meanwhile, a recent study has shown the significant differences in meibography grading between regional zones (nasal, central, temporal) and global grades [34]. This shows the need for a method able to extract, present and utilize morphological changes of meibomian gland structure on the local scale. Trying to meet this requirement, in this work an approach based on 2D Short Time Fourier Transform (2D STFT) is proposed [35]. The main advantage of this method is the application of 2DFT on small fragments of the Meibomian image, thus obtaining local values of the six chosen morphometric parameters. The 2D plots of those values (maps) provide an excellent tool for qualitative and quantitative description of the gland pattern. This additional information may help clinicians by highlighting the features of each morphometric parameter separately. A possible way of defining new morphological meibo-scores on the basis of the obtained intrinsic images is shown. As a final step we propose an introduction of new meibo-scores calculated from the morphometric images.

## Materials and methods

### Subjects

Subjects were healthy volunteers recruited from Faculty of Physics Adam Mickiewicz University in Poznan in Poland. Ethics clearance was issued by the institutional review board of

Adam Mickiewicz University of Poznan and adhered to the tenets of the Declaration of Helsinki. Before enrolment into the study all participants were informed about procedures used in the experiment. Written informed consent was obtained from all subjects.

The exclusion criteria were ocular allergies, eyelid and ocular surface disorders, recent ocular infections, any history of ocular surgery or continuous eye drop use. The 55 participants were contact lens wearers. Participants followed the recommendation not to wear contact lenses on a day before the examination procedure.

## Meibographic images

The Meibomian gland image analysis developed in this work was tested on the images acquired in our recent research [33]. A total of 146 images (2 images for both upper eyelids of each patient) were collected using home-built meibographic imaging equipment (details in the in the S1 Appendix). The aim was to provide a non-contact and patient-friendly acquisition method, preferably similar to other commercially available imaging techniques. Thus, the meibography system was mounted on the Topcon SL-D701 slit lamp which allowed to record meibographic images during the routine eye examination. An exemplary Meibomian image acquired with this device is presented in Fig 1.

In the present study only the images of the upper eyelid were collected for further analysis. The upper eyelids of the patients were everted to expose the embedded Meibomian glands and then a series of several images was acquired. The image of the best quality was selected as a representative for a given patient. Recorded photographs of the meibomian gland area were first preprocessed in *ImageJ* software. A set of filters was applied to firstly enhance the contrast (Fig 1b) and to eventually produce a binary version of the image showing only clear silhouettes of glands (Fig 1c). Then, the region of eyelid with Meibomian glands was manually marked (green dashed line in Fig 1b).

The recorded photographs were also subjectively graded by one experienced optometrist based on their distortions and then grouped into three categories: healthy (24 items), intermediate (75 items), unhealthy (47 items). This subjective analysis was based on several features of the gland pattern: gland direction, gland dilation, cut-off and narrowing [12, 19, 36, 37]. For example, the pattern was considered as distorted when its direction deviates from eyelid axis by more than 45˚. Meibographs were graded from 1 to 3 using following rule: no distortion of the Meibomian glands (healthy–grade 1); 1–4 Meibomian glands with distortion (intermediate–grade 2); more than five Meibomian glands with distortion (unhealthy–grade 3). Grading of the images was repeated on the following day. If the grades assigned on the two days were different, the images were reanalysed again to make a proper decision based on the presented criteria. Correlations between the evaluation results were estimated at $p < 0.05$ and $r = 0.794$. Other ocular symptoms and signs of the dry eye were not collected. The resulting classification served as a ground-truth standard for comparison with the outcome of the proposed automatic classification routine.

Two relevant morphological features of the gland pattern are shown in Fig 1a. The positions on the eyelid where the angle of deviation exceeds 45˚ are marked with red squares, whereas the regions with noticeable narrowing of glands are indicated with blue squares.

## Image analysis with 2D Fourier Transform

The use of the 2D Fourier transform (2D FT) method for Meibomian image analysis is justified by the observation that healthy Meibomian glands forms a periodic stripe pattern, whereas in the image of the glands described as unhealthy this pattern is often distorted [12, 24–25, 27]. The result of the 2D FT operation applied to different gland structures is schematically

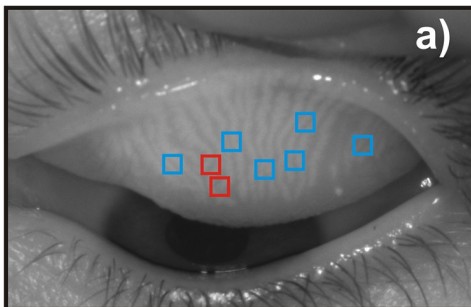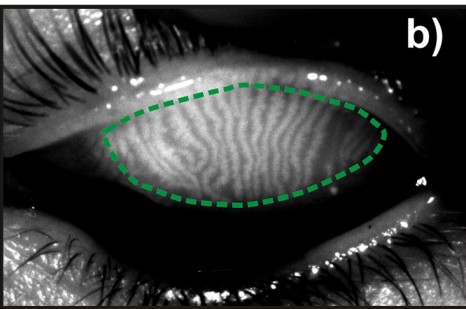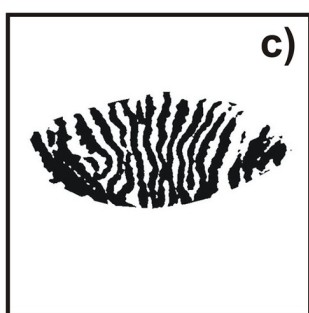

**Fig 1. Pre-processing and grading of Meibomian images.** a) original meibogram. Red squares show the positions where an angle of deviation exceeds 45˚. Blue squares indicate regions with noticeable narrowing of glands. b) Meibomian image with increased contrast. The green dashed line marks the area of eyelid with Meibomian glands. c) binarized Meibomian image.

presented in Fig 2. If the analyzed image shows a unidirectional gland structure with a constant width and a constant distance between the glands (which corresponds to a well-defined spatial frequency), then a pair of characteristic sharp peaks appear in the Fourier-transformed image (Power Spectral Density, PSD, Image) with the center of coordinate system as the center of symmetry (Fig 2). Their distance from the center of the PSD image is a measure of the spatial frequency (corresponding to gland width or separation), while their orientation corresponds to the direction of the gland structure. As illustrated in Fig 2, the change in the gland pattern orientation and in the width of the glands results in corresponding characteristic changes in the PSD image. Real Meibomian gland structures are never perfect and there is always a distribution in gland's width or separation, as well as in their orientation. As a result, broadening of the spectral features in PSD images occurs. Therefore, the information on gland's distortion is encoded in the shape of the spectral features of PSD image.

### Determination of intrinsic images with 2D Short-Time Fourier Transform

To determine the morphological properties locally, the method utilizes the so called Short-Time Fourier Transform (STFT) in a manner similar to that used previously to enhance the analysis of fingerprint images [38]. Application of 2D SFTF applied to the real Meibomian image (Fig 1) is shown in Fig 3. The analyzed image (Fig 3a) is divided into smaller regions and the 2D FT transformation is performed for each region separately. The regions are selected by a window of a given shape and position (Fig 3b and 3e). In order to achieve a uniform map of calculated parameters, the window position was assigned to every 10th pixel of the original Meibomian image (blue dots in Fig 3a). Details of the 2D STFT analysis are provided in the S2 and S3 Appendices.

The PSD image represents the distribution of spatial frequencies along (*x,y*) coordinates of the original image (Fig 3c and 3f). In this representation, the distance ($q$) from the center of the Fourier transformed image is a measure of the spatial frequency of the gland pattern (related to gland width), whereas the angle ($\theta$) is connected to the orientation of the gland pattern. The PSD image was transformed from cartesian ($q_x$, $q_y$) to polar coordinates ($q$, $\theta$) (Fig 3d and 3g). The advantage of this operation is that PSD($q$, $\theta$) can be interpreted as distribution of probability $p(q, \theta)$ for gland features of a given frequency, $q$, and orientation, $\theta$, existing in the analyzed region of Meibomian image. Marginal density function $p(q)$ and $p(\theta)$ calculated from $p(q, \theta)$ (S2 Appendix) were then compared with theoretical models. The result of this procedure is presented on Fig 4.

**Fig 2. Schematic illustration of 2D Fourier Transformation.** Transformation of an image of undistorted gland pattern results in a PSD image showing two sharp peaks of well-defined position and orientation (black circles in bottom images). Orientation and separation of these peaks correspond to direction and frequency of the gland pattern, respectively. When the gland pattern is not uniform, some distribution in frequency and orientation occur. As a result, the spectral features in PSD images tend to smear out.

As follows from Fig 4, experimental $p(q)$ and $p(\theta)$ distributions (open symbols) show a clear peaks localized at certain positions and characterized by their width. In order to parametrize these features, an assumption was made that gland frequency, $q$, and orientation, $\theta$, are random variables described by normal distributions and Gaussian and von Mises [39] distributions (S4 Appendix) were used to fit experimental $p(q)$ and $p(\theta)$, respectively (solid lines). The obtained values of the peak positions ($q_0$ and $\theta_0$) correspond to the mean values of gland frequency and orientation, respectively, whereas peak widths (parametrized by variances $\sigma_{q,fit}$ and $\sigma_{\theta,fit}$) represent uncertainties in estimation of these parameters. Interpretation of such obtained variances needs some caution. As $p(q)$ and $p(\theta)$ distributions were obtained from a Fourier transform of a windowed image, the widths of these distributions are naturally broadened resulting from a finite size of the window (see S3 Appendix). The ideal $p(q)$ and $p(\theta)$ distributions, expected for perfect gland structure (characterized by a constant frequency and constant orientation), are shown in Fig 4 as shaded areas. These distributions will be widened if the real gland image differs from the ideal one (as the gland distortion increases). For such a case, $\Delta\sigma$ parameter (being the difference between the widths of real and ideal distributions) will take a finite (non-zero) value.

As the real Meibomian gland pattern always shows distortions, the shapes of corresponding probability distributions will change across the eyelid and will depend on the window position (Fig 2). The $p(q)$ and $p(\theta)$ distributions were then determined for different window positions so to cover the entire Meibomian gland image (blue dots in Fig 3a). For every window position the values of four parameters (namely: $q_0$, $\Delta\sigma_q$, $\theta_0$, $\Delta\sigma_\theta$) were directly extracted from probability distributions. Plotting the color-coded values of these parameters as a function of window position produces four maps being spatial distributions of individual morphometric parameters' values. The images of gland frequency, $q_0$, and gland orientation, $\theta_0$, were used to generate two additional maps, namely the map of frequency gradient, $G_q$, and the map of angular

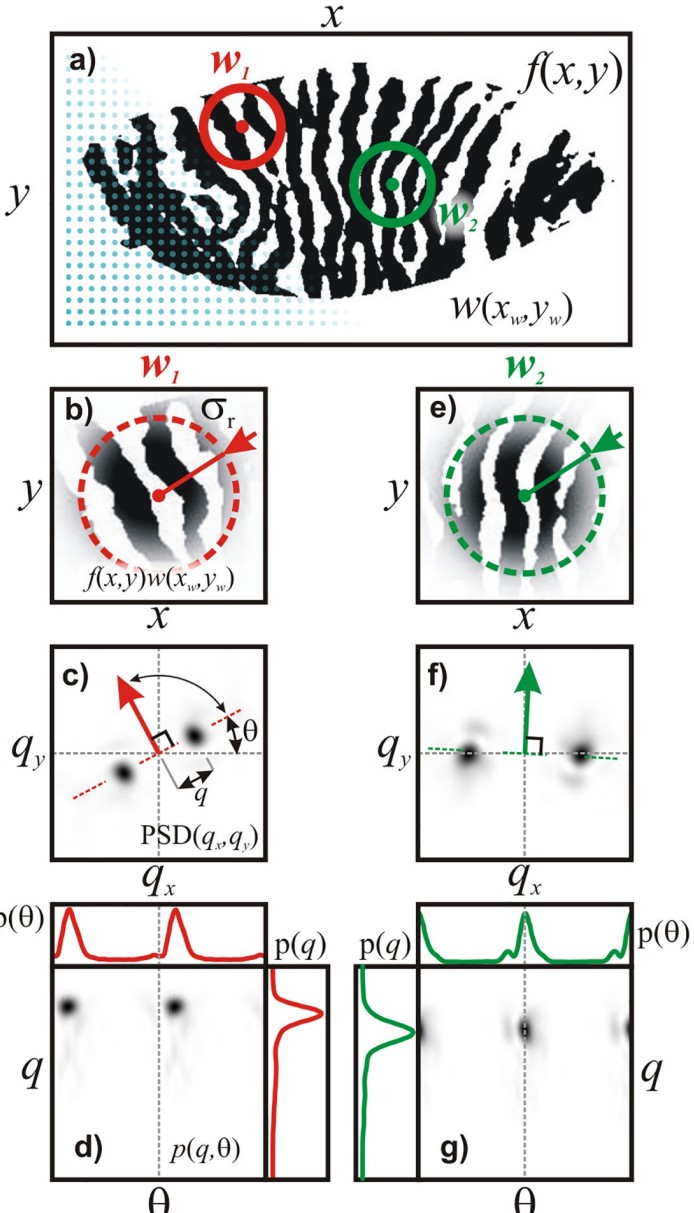

**Fig 3. 2D Short-Time Fourier Transform (STFT) used for determination of local probability density for gland frequency, $p(q)$ and orientation, $p(\theta)$.** Panel a) shows original (binarized) Meibomian gland image, $f(x,y)$. During the STFT analysis of an image, the Gaussian window is placed in strictly defined positions $(x_w, y_w)$, which are illustrated a grid of blue dots. Two arbitrary positions of the window, $w_1$ and $w_2$, are presented as a red and a green circle, respectively. Panels b) and e) shows a Meibomian image limited by windows $w_1$ and $w_2$, respectively. The radius of the dashed circles show the width (variance) of the Gaussian window, $\sigma_r$. The image limited by $w_1$ shows broader and more inclined gland structure then that seen in $w_2$. Panels c) and f) shows PSD in Cartesian coordinates calculated for images limited by windows $w_1$ and $w_2$, respectively. The dark spots are the spectral features whose radial distance, $q$, informs about gland pattern spatial frequency, whereas the angular distance, $\theta$, corresponds to the gland orientation. The thick colored arrow (pointing at $\theta$+90° direction) indicates the mean gland orientation. Notice how it complies with the real structure shown in panels b) and e). Panels d) and g) show PSD in polar coordinates calculated for images limited by windows $w_1$ and $w_2$, respectively. In this representation PSD corresponds to a probability map of finding a gland structure with a given frequency, $q$, and orientation, $\theta$. Marginal plots show local probability distributions $p(q)$ and $p(\theta)$ which were obtained by projection of $p(q, \theta)$ on the appropriate axis (S2 Appendix).

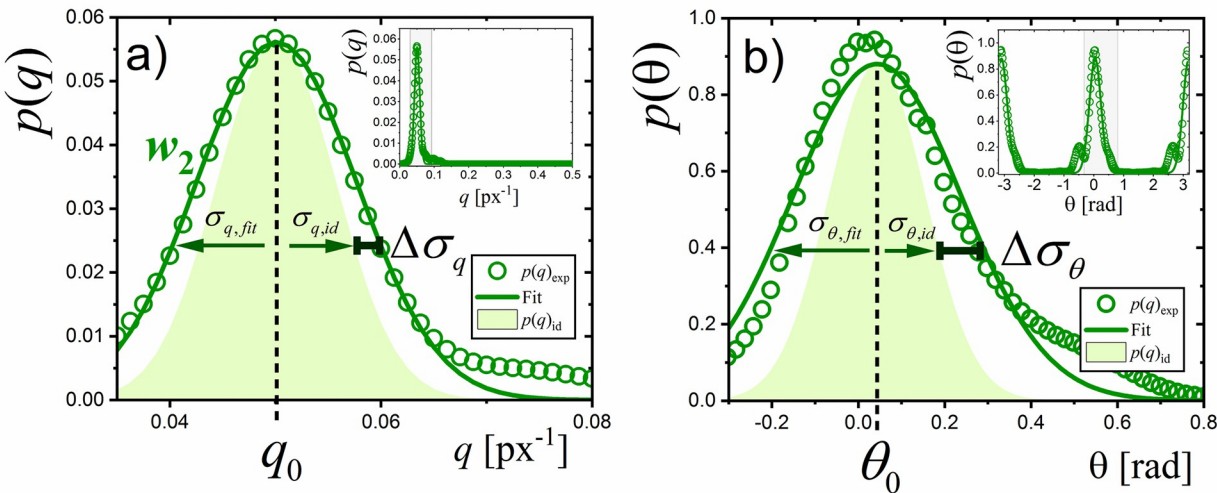

**Fig 4. Derivation of morphometric parameters ($q_0$, $\sigma_q$, $\theta_0$, $\sigma_\theta$) by the analysis of local probability distributions for a single window position (window $w_2$ from Fig 3).** Panel a) shows the $p(q)$ distribution indicating a probability of finding a gland structure with a given spatial frequency, $q$. Open circles are experimental data. Solid line is the fitting result with normal distribution providing the values of maximum, $q_0$, and the variance, $\sigma_{q,fit}$. The maximum of the distributions is a measure of the local gland frequency, $q_0$. Shaded area shows the *ideal* distribution with a variance, $\sigma_{q,id}$, expected for a constant frequency gland pattern limited by a Gaussian window. Broadening of the experimental distribution with respect to the ideal one, $\Delta\sigma_q$, normalized to the $\sigma_{q,id}$, is a measure of true gland frequency variance, $\sigma_q$. The inset in (a) shows the $p(q)$ in the whole range of $q$ values. Panel b) shows the $p(\theta)$ distribution indicating a probability of finding a gland pattern with a given orientation, $\theta$. Open circles are experimental data. Solid line is the fitting result with von Mises distribution providing the values of maximum, $\theta_0$, and the variance, $\sigma_{\theta,fit}$. The maximum of the distributions is a measure of the local gland orientation, $\theta_0$. The shaded area shows the *ideal* distribution with a variance, $\sigma_{\theta,id}$, expected for a constant frequency gland structure limited by a Gaussian window. Broadening of the experimental distribution with respect to the ideal one, $\Delta\sigma_\theta$, normalized to the $\sigma_{q,id}$, is a measure of true gland frequency variance, $\sigma_\theta$. The inset in (b) shows the $p(\theta)$ in whole range of $\theta$ values.

incoherence, $C_\theta$ (details are given in S4 Appendix). Therefore, as a result of 2D STFT analysis are six morphometric maps are generated from a single Meibomian image.

## Results

Fig 5 shows the result of a 2D Short Time Fourier Transform 2D STFT analysis performed on three exemplary Meibomian images belonging to different categories. A direct comparison of the original images (Fig 5 row a) shows that well-defined unidirectional stripe pattern characteristic for healthy glands gradually disappears with an ailment progression. For the 'Unhealthy" case it is possible to identify specific regions of the image where the gland width (or separation) clearly changes, as well as areas where glands obviously change their direction. In reality, changes in the width and orientation of the glands may be more subtle, sometimes even difficult to see, and occur over the entire eyelid area where regions of varying width and orientation interpenetrate each other. With a 2D STFT analysis these various contributions were disentangled to create separate images (morphometric maps), each showing spatial distribution of only one of morphological parameter of the gland pattern. These intrinsic images are shown in Fig 5b–5g.

As follows from Fig 5, the corresponding maps calculated from images representing various glands condition clearly differ from each other and reflect the gland's condition. In order to parametrize these changes, for each parameter, the distribution of its values was determined (Fig 6) and the shapes of the distributions were quantified with five measures of distribution, namely: Entropy, Mean, Variance, Skewness and Kurtosis (S5 Appendix). This gives in total 30 descriptive features for each Meibomian image.

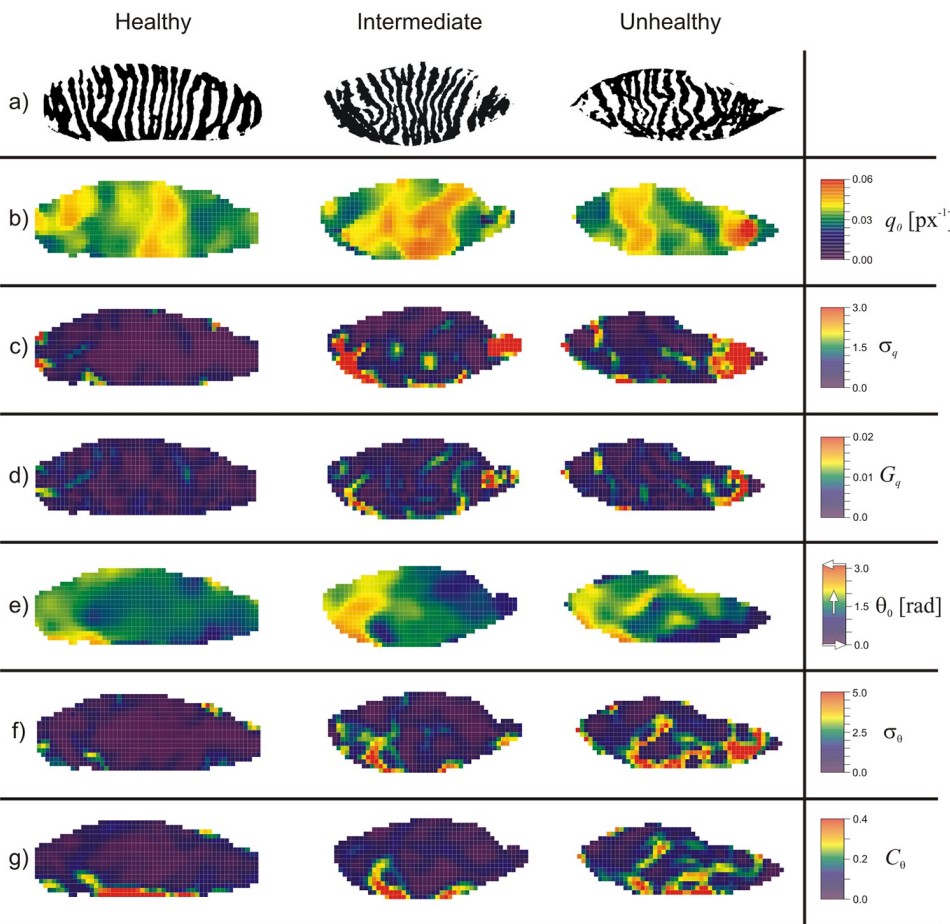

**Fig 5. Six sets of morphometric maps calculated from three Meibomian images classified as healthy, intermediate and unhealthy (in columns).** Subsequent rows show: a) original (binarized) Meibomian images; b) maps of gland frequency, $q_0$; c) maps of gland frequency variance, $\sigma_q$; d) maps of frequency gradient, $G_q$.; e) maps of gland orientation, $\theta_0$; f) maps of gland orientation variance, $\sigma_\theta$; g) maps of angular incoherence, $C_\theta$.

Using principal component analysis (PCA) and linear discriminant analysis (LDA) [40] the dimensionality of the dataset was reduced from 30 features to only 2 new variables which best describe the data: $PCA_{1,2}$ (or $LDA_{1,2}$). The correlation plots of both $PCA_{1,2}$ (and $LDA_{1,2}$) components extracted for all meibomian images are shown on Fig 7. Marginal plots on Fig 7 were interpreted as probability distributions of corresponding components ($PCA_{1,2}$ or $LDA_{1,2}$).

Knowing the probability distributions of a given component ($PCA_{1,2}$ or $LDA_{1,2}$) for each category, a simple threshold classifiers were created: an image being parametrized with a pair of component values ($PCA_1/PCA_2$ or $LDA_1/LDA_2$) is assigned to a category specified by the highest value of the product of corresponding probability functions ($p(PCA_1)p(PCA_2)$ or $p(LDA_1)p(LDA_2)$). The classification performance of this approach is presented in Table 1. For more information see S6 Appendix.

## Discussion

Fig 1 shows how the morphological condition of the glands are "traditionally" assessed. Although more descriptive features have been defined in the literature [19–20, 24–28], the following discussion focuses on only two examples: the angle of deflection and the narrowing of

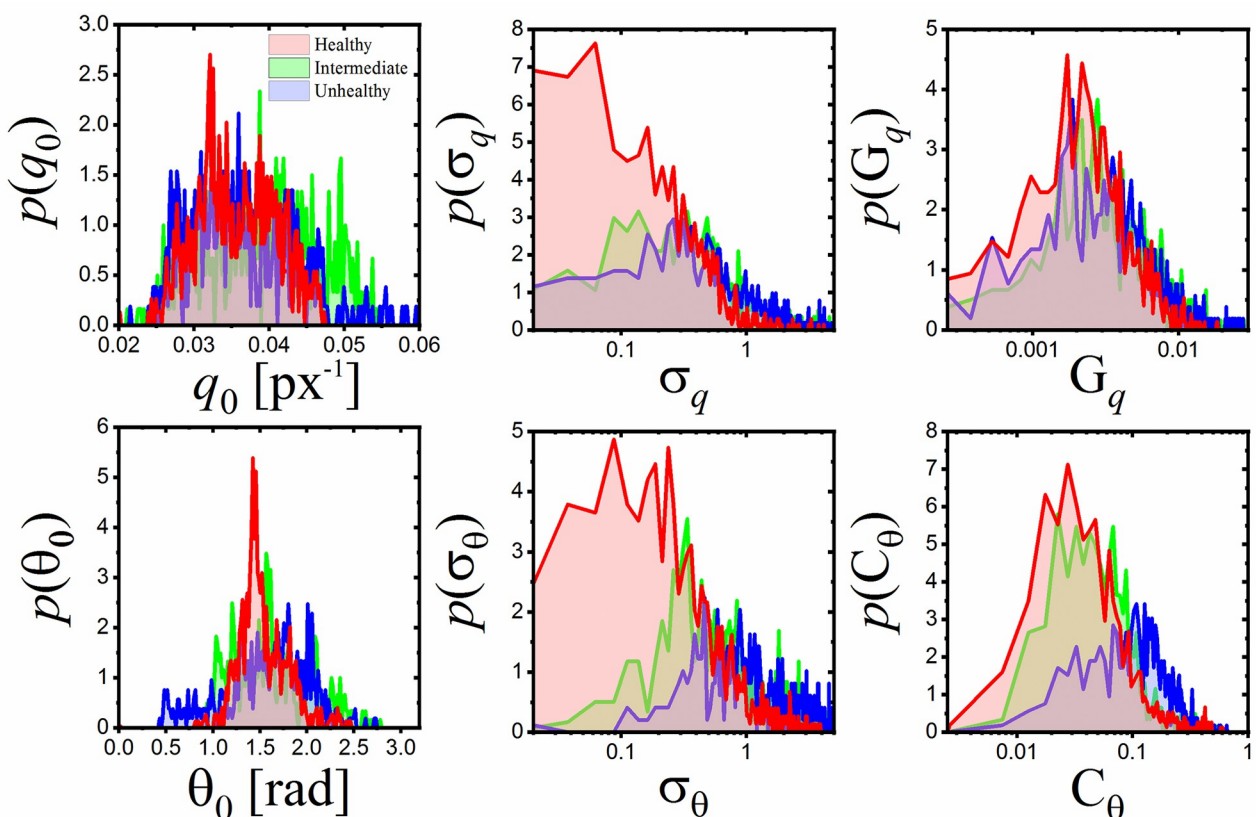

**Fig 6. Distributions of pixel values for 6 morphometric maps plotted for Meibomian images classified as healthy, intermediate and unhealthy.** Notice that the shape of each distribution depends on the image category (gland ailment). To quantify these changes five measures of distribution were calculated (Entropy, Mean, Variance, Skewness and Kurtosis) giving in total 30 descriptive features for each Meibomian image.

the glands. From Fig 1 it is clear that both these features occur in different and separate locations on the eyelid surface. Moreover, there are regions were glands shows clear angular deviation but the angle value is smaller than the arbitrary threshold value (45° in this case). Similarly, there are many locations where glands obviously narrow (or broadens), however as the "narrowing" of glands is not well defined, it is hard to pinpoint those regions. It is clear that any valuation protocol being based on comparing the value of a certain morphological feature with an accepted threshold value will focus only on very specific regions of an eyelid. Regions other than those will be omitted in assessment procedure and treated as clinically irrelevant. Lastly, the clinical description of a single Meibomian image with only two morphological features requires many annotations and calculations and, because of arbitrariness in the feature definition, is very subjective.

The method proposed in this work allows for overcoming the difficulties mentioned above by automatic mapping of objectively determined values of different morphometric properties across entire eyelid surface. The results of such analysis are presented in the form of 6 morphometric maps, as in Fig 5.

One of the most obvious morphological feature of Meibomian gland pattern is their width or their separation. The presented image analysis method estimates this property using the gland pattern frequency, $q_0$ (being an inverse of gland width and/or gland-gland separation) and visualizes this property on the entire surface of the eyelid. Comparing original gland structures (row a in Fig 5) with corresponding maps of gland frequency $q_0$ (row b on Fig 5), it is

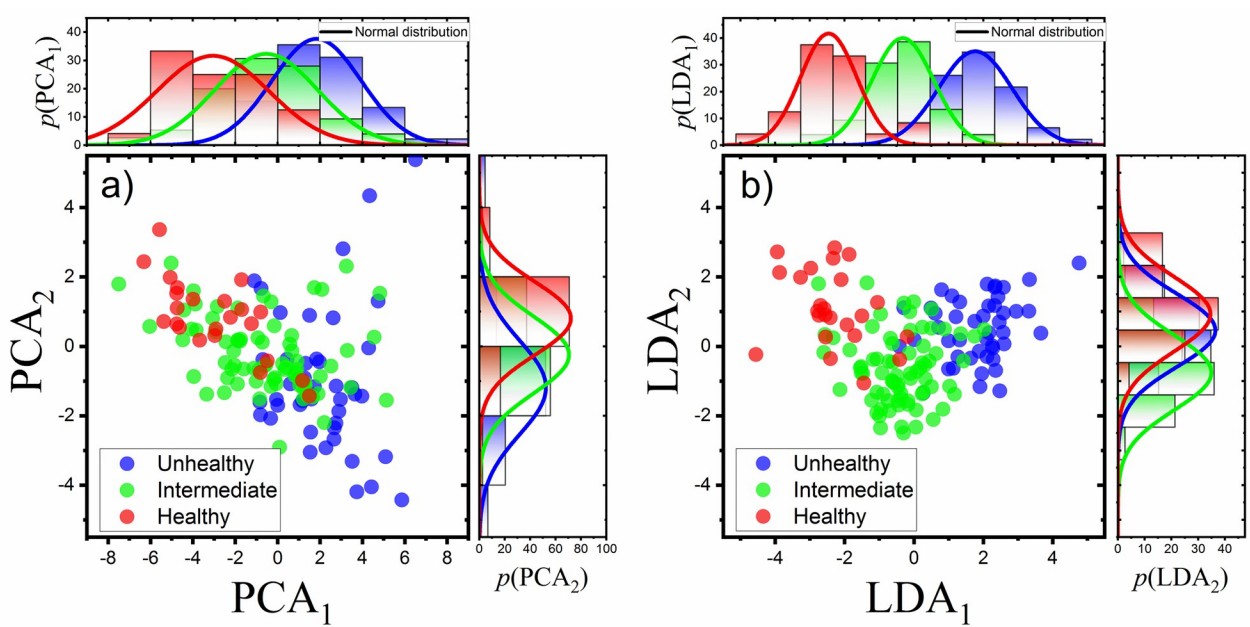

**Fig 7. Correlation plots between first two components of a) Principal Component Analysis (PCA) and b) Linear Discriminant Analysis (LDA) for images classified as healthy, intermediate and unhealthy.** Marginal plots show probability distributions of corresponding components. Notice that although both analyses provide noticeable separation of categories, the LDA analysis (being a supervised method) provides much better clustering of classes.

clear that the regions with higher values of $q_0$ correspond to regions where narrower glands are observed. Hence, the map of gland frequency $q_0$ allows for easy identification of glands narrowing regions.

If the gland separation changes within the window of STFT analysis, then the value of $q_0$ is estimated with some uncertainty. This property is shown on the map of gland pattern frequency variance, $\sigma_q$ (row c of Fig 5). Areas with low $\sigma_q$ values correspond to a gland pattern with well-defined value of frequency (well-defined gland width or well-defined gland separation).

For better localization of areas in which narrowing or broadening of Meibomian glands occurs, it is helpful to determine how quickly the glands are changing their width (or separation). The map of frequency gradient, $G_q$ (Fig 5d), shows the rate of change in the gland pattern frequency. Looking at this map one notices that areas where gland narrowing or broadening occurs are clearly highlighted, whereas regions where the frequency of gland structure does not change much are mapped with low value of $G_q$.

As mentioned earlier (Fig 1), a common method of assessing the glands morphology is based on the absolute values of the gland's angle of deflection and involves counting the events of exceeding a certain threshold angle value (45°). This task can be facilitated by determining

**Table 1. Classification efficiency (in percentage of correctly classified) for different classifiers (column headers) used to distinguish images of Meibomian glands.**
95% Confidence Interval (see S7 Appendix for details).

| Classifier | Healthy | Intermediate | Unhealthy |
|---|---|---|---|
| PCA | 83±12 | 48±4 | 83±9 |
| LDA | 88±14 | 79±9 | 91±12 |
| PCA [33] | 88±11 | 46±3 | 83±10 |

an exact value of glands angle for every position of an eyelid. This is exactly what is presented on the map of gland orientation, $\theta_0$ (Fig 5e).

The values of deviation angle, $\theta_0$, are estimated with some uncertainty. The measure of this uncertainty is shown on the map of gland orientation variance, $\sigma_\theta$ (Fig 5f). Similarly to $\sigma_q$ map, regions with low values of $\sigma_\theta$ correspond to gland pattern with well-defined orientation. We recall that this parameter but measured on the global scale (for the whole Meibomian image), was used previously as a measure of anisotropy in gland periodicity [33].

Aside from a knowledge of the glands angle at certain location, it may be just as important to visualize where this angle is changing. This property is shown on the map of angular incoherence, $C_\theta$ (Fig 5f), which shows the spatial variation in the mean direction of gland pattern. Comparing original Meibomian images (row a in Fig 5) with corresponding maps of $C_\theta$, one can easily notice that regions where the orientation of glands suddenly changes are highlighted. Regions with low value of $C_\theta$ correspond to locations where glands are orientated in roughly similar direction.

It is worth recalling that the morphometric maps are calculated directly from a raw Meibomian image. Therefore, if the described approach was implemented in the meibograph control software, the clinician would have access to them immediately after taking a picture of the glands. Thanks to the large amount of objective information collected in the form of morphometric maps, qualitative analysis of meibomian gland morphological condition is made easier. Even simple visual inspection of the maps presented in Fig 5 may be useful in clinical practice and may improve the accuracy of the diagnosis.

It is also possible to attempt a more advanced analysis of the obtained results. Because the presented method produces new images, a further image analysis can be performed on each of them. For example, similarly to the popular drop-out area parameter (the ratio between the area occupied by Meibomian glands to the total area of the eyelid), it is possible to define simple measures of gland deformity by comparing the areas occupied by glands considered to be deformed to the total area of the eyelid. This measure can easily be obtained by firstly comparing the pixel numbers present in the appropriate morphometric map from Fig 5, with a certain value considered as the threshold between undisturbed and distorted state. Then the value of a new morphometric parameter is calculated as the ratio between the number of pixels that exceed the threshold value to the total number of pixels in the map. The effect of such a procedure performed on the image of frequency gradient, $G_q$ (Fig 5d) and on the image of angular incoherence, $C_\theta$ (Fig 5f) is presented in Fig 8. Using these particular images, the values of two morphometric measures were estimated: 1) $A_q$ quantifies the percentage of an eyelid area where significant changes in the glands width occur (that is narrowing and broadening); 2) $A_\theta$ measures the percentage of an eyelid area where the change in glands orientation is noticed.

Fig 8 clearly shows that the area covered by deformed Meibomian glands correlate with the ailment progression. In an Meibomian image classified as unhealthy, morphological changes concern a significant area of the eyelid and meiboscores $A_q$ and $A_\theta$ take correspondingly high values. The surface of healthy glands is much less affected which results in lower values of $A_q$ and $A_\theta$. Interestingly, the regions indicated by the human specialist and considered to be disturbed coincide roughly with the areas highlighted automatically with the presented method (compare Fig 1a with the Fig 8b/Intermediate). Looking at the values of $A_q$ and $A_\theta$ it is also clear that some of them are similar even if estimated for images belonging to different categories. This shows that a correct morphological condition assessment should not be based on a single parameter only.

The above-estimated morphometric scores ($A_q$ and $A_\theta$) are only an example of the possibility offered by the presented method. On the basis of morphometric maps (Fig 5), other measures of the morphological state can be defined just as easily. This makes the proposed

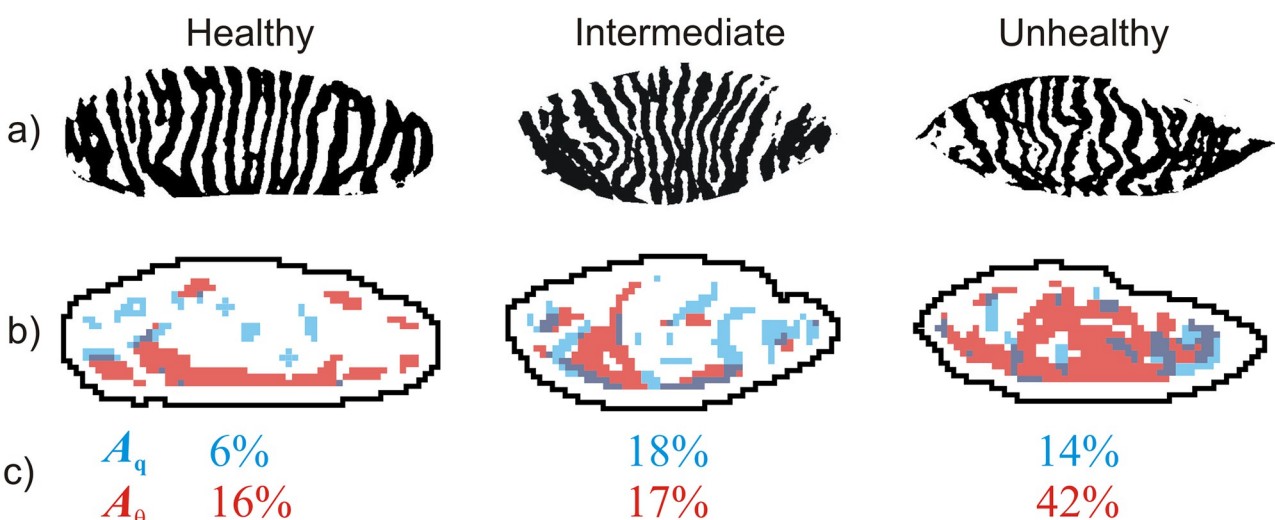

**Fig 8. Definition of exemplary new morphometric parameters using the maps presented in Fig 5.** Row a) original Meibomian images; Row b) locations where frequency gradient (blue regions) or angular incoherence (red regions) exceed an arbitrary threshold values of 0.007 and 0.1, respectively; Row c) percentages of the areas covered by glands with significant changes in the glands width ($A_q$, are of blue region) and with noticeable change in glands orientation ($A_\theta$, area of red region).

approach very promising as there is a strong evidence that the Meibomian gland morphological changes correlate with the ocular symptoms and signs of the dry eye [10–12]. The further work should focus on the associations between the obtained local morphometric features and the ocular symptoms and signs of the dry eye disease, which may confirm the clinical usefulness of the proposed approach.

In addition to the above detailed analysis of individual morphometric maps, it is interesting to check whether the mapped values of morphometric parameters can be used for the automatic classification of images. This will serve as an additional indirectly confirmation that the morphometric maps contain clinically useful information. As follows from Fig 5, for healthy Meibomian glands, the morphological properties sensitive to homogeneity in the frequency and in the orientation of Meibomian glands ($\sigma_q$, $G_q$, $\sigma_\theta$ and $C_\theta$) take very low values and their maps are rather uniform. However, when gland ailment progresses, the shape of the glands begins to distort in some place. As a result, the values of homogeneity-sensitive properties increase in the corresponding position of the image. This observation suggests that distribution of pixel values presented on intrinsic images should be different for different categories. For healthy images most pixel values are expected to be low. Distribution should move to higher pixel values when the ailment progresses. This situation is well illustrated on Fig 6 where pixel value distributions for each intrinsic image are presented. The shapes of these distributions were quantified by determining their Entropy, Mean, Variance, Skewness and Kurtosis. As a result, for each Meibomian image 30 descriptive features were determined (6 intrinsic images x 5 measures of distribution).

Differences in the values of the 30 descriptive features found for each Meibomian image can be used to automatically categorize the images into subjective categories. There are a number of machine learning algorithms that can be used for this purpose. Finding the best solution based on its classification efficiency is beyond the scope of the present work. Therefore, the performance of only one simple classifier was tested, which separates Meibomian images based on their probability of belonging to a certain category. The classification performance of this approach is presented in Table 1.

As follows from Table 1, both classification approaches give satisfactory results, although the PCA classifier performs worse than the LDA. This is especially true for the category "intermediate". This observation can be explained by broad and strongly overlapped probability distributions (marginal plots in Fig 7a) making the distinction between the classes inherently uncertain. Comparing current results from PCA classifier with the previous outcomes [33] one sees that increasing the number of descriptive features (current 30 features vs. previous 2) is not the way for improving categorization efficiency. The data reduction method based on maximizing variability in the data set (utilized by PCA approach) has probably hit the limit of efficiency. Further increase in categorization performance can be obtained using different algorithms. This is demonstrated by the output of the same type of classifier but using LDA data reduction method. As follows from Fig 7b, for LDA approach much better clustering of data points belonging to different categories was obtained. As a result, appropriate probability distributions are narrower and better separated which translates into observed classification improvement.

## Conclusions

The presented method for Meibomian image analysis allows for a truly objective estimation of few strictly defined morphometric parameters. The newly developed automated procedure calculates numerical values of these parameters and generates their maps across entire eyelid area thereby allows for tracking the local morphometric changes. Moreover, each map presenting particular morphological property can be subjected to further detailed analysis to extract even more quantitative information and define new morphometric scores. Isolating individual morphometric components from the original Meibomian image may help clinicians to see in which part of the eyelid disturbance is taking place and also to quantify it with a numerical value providing a better insight into disease pathophysiology. Since many ophthalmic disorders start with a slight deformation of the meibomian glands (before their atrophy begins) the results based on the presented method may be particularly important in detection of the initial stages of Meibomian gland disease.

Automatic categorization of Meibomian images was successfully performed confirming that the maps of morphological parameters contain clinically useful information and that taking into account more morphological features can improve classification efficiency.

The presented method is fast, user-friendly and can be integrated with Meibograph software. To confirm its clinical utility, further work should focus on the associations between the introduced morphometric parameters with the ocular symptoms and signs of the dry eye disease.

## Supporting information

**S1 Appendix. Meibomian gland images acquisition.**
(PDF)

**S2 Appendix. 2D Short-Time Fourier Transform.**
(PDF)

**S3 Appendix. The consequence of finite width of Gaussian window.**
(PDF)

**S4 Appendix. Calculation of intrinsic images.**
(PDF)

**S5 Appendix. Quantification of intrinsic images.**
(PDF)

**S6 Appendix. Dimensionality reduction and image categorization.**
(PDF)

**S7 Appendix. Estimation of confidence interval for classification efficiency.**
(PDF)

**S1 Raw images.**
(ZIP)

## Author Contributions

**Conceptualization:** Kamila Ciężar, Mikolaj Pochylski.

**Formal analysis:** Kamila Ciężar.

**Methodology:** Mikolaj Pochylski.

**Software:** Mikolaj Pochylski.

**Supervision:** Mikolaj Pochylski.

**Writing – original draft:** Mikolaj Pochylski.

**Writing – review & editing:** Kamila Ciężar, Mikolaj Pochylski.

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
