## [Decision Letter · Decision Letter 0]

28 Mar 2022

PONE-D-22-014302D Short-Time Fourier
Transform for local morphological analysis of meibomian gland
imagesPLOS ONE

Dear Dr. Pochylski,

Thank you for submitting your manuscript to PLOS ONE. After careful consideration, we
feel that it has merit but does not fully meet PLOS ONE’s publication criteria as it
currently stands. Therefore, we invite you to submit a revised version of the
manuscript that addresses the points raised during the review process.

 The study provides an fourier transform image analysis
approach for images of patient upper eyelid meibomian glands, acquired using a
custom-built meibographer. The approach is potentially useful especially if it can
be combined with patient signs and symptoms, but further details are required.
Please also address all the comments from both reviewers listed
below. 1. Please include
ethics approval details for the current study. This used images from the study in
The Ocular Surface 2020 (also not very details regarding ethics). Details on
inclusion/exclusion criteria should also be included, and how many patients were
examined should be stated. The number of images is noted in the manuscript, and that
there were 2 images per person, but 'n' not
provided. 2. Please give
full details on the imaging techniques used for meibography. The authors used a
custom built meibographer (based on the2020 study). How does this compare to the
clinically used (commercial)  instruments, in terms of image resolution for of upper
eyelid meibomian glands. Have the authors used their 2D fourier transform analysis
on routine clinical meibography
images? 3. Can the
individual meibomian gland analysis described here be practically applied in a
clinical context - the analysis is complex and it s not clear how this would be
incorporated into a routine clinical
examination. This is
important to justify, noting the conclusion (line 316): "To conclude, our method for
isolating individual morphometric components from the original Meibomian image can
be of invaluable aid in the diagnostic process. It may help clinicians to see in
which part of the eyelid disturbance is taking place and also to quantify it with a
numerical value providing essentially insight into disease
pathophysiology." The
results presented do not really support the 'invaluable aid' comment and should be
modified as such, 4. Can
the authors comment on associations between the eyelid meibography 'heatmaps'
generated using the local analysis, and symptoms reproted by the patients. Thank
you. 5. A careful edit for
English grammar and expression in the entire manuscript would improve the
communication; there are sections that are not well
expressed. Please address
all the comments provided by the reviewers

Please submit your revised manuscript by May 12 2022 11:59PM. If you will need more
time than this to complete your revisions, please reply to this message or contact
the journal office at plosone@plos.org. When
you're ready to submit your revision, log on to https://www.editorialmanager.com/pone/ and select the 'Submissions
Needing Revision' folder to locate your manuscript file.

Please include the following items when submitting your revised
manuscript:A rebuttal letter that responds to each point raised by the academic
editor and reviewer(s). You should upload this letter as a separate file
labeled 'Response to Reviewers'.A marked-up copy of your manuscript that highlights changes made to the
original version. You should upload this as a separate file labeled
'Revised Manuscript with Track Changes'.An unmarked version of your revised paper without tracked changes. You
should upload this as a separate file labeled 'Manuscript'.

If you would like to make changes to your financial disclosure, please include your
updated statement in your cover letter. Guidelines for resubmitting your figure
files are available below the reviewer comments at the end of this letter.

We look forward to receiving your revised manuscript.

Kind regards,

Michele Madigan

Academic Editor

PLOS ONE

Journal Requirements:

2. Please include an ethics statement in your Methods section for how the images were
originally obtained. Please also amend your Data availability statement in the
submission form to declare how others may obtain access to these images.

Reviewers' comments:

Reviewer's Responses to Questions

**Comments to the Author**

1. Is the manuscript technically sound, and do the data support the conclusions?

Reviewer #1: Partly

Reviewer #2: Partly

2. Has the statistical analysis been performed
appropriately and rigorously? 

Reviewer #1: No

Reviewer #2: Yes

3. Have the authors made all data underlying the
findings in their manuscript fully available?

Reviewer #1: Yes

Reviewer #2: Yes

4. Is the manuscript presented in an intelligible
fashion and written in standard English?

Reviewer #1: Yes

Reviewer #2: No

5. Review Comments to the Author

Reviewer #1: This manuscript describes a novel image analysis method to provide
detailed information on the local morphology of meibomian glands. This was then
tested on meibomian gland images which were classified as healthy, intermediate and
unhealthy. The bulk of the manuscript seems to contain a very detailed and technical
explanation of the development of the image analysis technique, which to be honest,
as a clinical researcher, I found challenging to follow. My general comments
include:

• Please include details of the original ethics approval, from which the images were
obtained. It would also be helpful to understand the general inclusion/exclusion
criteria of the original study population

• Materials and Methods – the meibomian gland images were graded 1-3 on two
consecutive days and the results were well correlated. However, it would be helpful
if the authors could clarify how many researchers were involved in the grading, and
how the images were classified if the grades assigned were different on the two
days

• Table 1 – please add confidence intervals or other statistical analyses to this
table. Otherwise, it is difficult to determine whether these proportions are
significantly different from one another

• Line 211 belongs in the Materials and Methods section, but further detail needs to
be added regarding the “home-built meibographic imaging equipment” used. How does
this compare to commercially available imaging tools?

• Line 121 – the sentence beginning “In next section” is unnecessary

• The manuscript should remove all personal pronouns e.g. line 121 “we”, line 132
“our”

• Line 316 – please soften the conclusion “invaluable aid” – this is too strong and
not necessarily supported by the data

• Do the findings correlate with any ocular symptoms or other signs of dry eye?

• What is the clinical utility of this analysis system? Is it something that can be
easily incorporated into clinical practice? What do the findings actually mean,
since no associations are discussed relating the health of the glands and other
signs and symptoms of dry eye?

Reviewer #2: How was MG function scored?

How user-friendly this method is?

Adding 1 or 2 sample images to the manuscript will help readers better understand and
visualize to what extent authors are referring to deformity of glands.

Also, I would like to know about the complete tarsal area what was exposed to
calculate MGs height. Height of MGs may differ with eversion specially for lower
lids as we have less surface area.

Good but do comment on use fullness of this method. What is the clinical utility of
this technique?

Tortuosity is related more to the twisting or curved MGs. In literature, there are
several methods to measure tortuosity and deformity. I would like to know more about
the method authors have used and its reliability as well. Better to describe with
sample images.

There are various ways to quantify Meibomian gland images particularly
three-dimensional Fourier domain OCT. How this 2D Fourier transform is better?

Better to revise keeping in mind above mentioned points.

6. PLOS authors have the option to publish the peer
review history of their article (what does this mean?). If published, this will
include your full peer review and any attached files.

If you choose “no”, your identity will remain anonymous but your review may still be
made public.

**Do you want your identity to be public for this peer review?** For
information about this choice, including consent withdrawal, please see our
Privacy Policy.

Reviewer #1: No

Reviewer #2: No

---

## [Author Response · Author response to Decision Letter 0]

10 May 2022

Response to the points raised by an Academic Editor:

"1. Please include ethics approval details for the current study. This used images
from the study in The Ocular Surface 2020 (also not very details regarding ethics).
Details on inclusion/exclusion criteria should also be included, and how many
patients were examined should be stated. The number of images is noted in the
manuscript, and that there were 2 images per person, but 'n' not provided."

New subsection named “Subjects” in “Materials and methods” was introduced where
details on ethics approval, inclusion/exclusion criteria and number of examined
patients were included.

"2. Please give full details on the imaging techniques used for meibography. The
authors used a custom built meibographer (based on the 2020 study). How does this
compare to the clinically used (commercial) instruments, in terms of image
resolution for of upper eyelid meibomian glands. Have the authors used their 2D
Fourier transform analysis on routine clinical meibography images?"

Standard IR retro-reflection meibograph is very simple device. Actually it is
ordinary camera adapted to work with infrared light. The quality of the acquired
images is determined just by the quality of the lens, resolution and dynamical range
of the camera. The parameters of our system (resolution 20pix/mm and 8-bit dynamics)
were sufficient to record a clear image of the basic structure of the Meibomian
glands, that is, the feature we were interested during our study. We added new
figure (Fig.1) which shows an exemplary raw meibographic image acquired by our
system. As one can see it does not look much different from what is obtained with
commercial instruments. We did not test our method on images acquired with such
devices. It is certain that the quality of the images provided by these instruments
will not be worse than ours. Therefore, we are confident that the described method
will work also for that case. 

More information on imaging technique used for meibography was introduced into
“Meibographic images” section of “Materials and methods” as well as in dedicated “S1
Appendix. Meibomian gland images acquisition” supplementary document.

"3. Can the individual meibomian gland analysis described here be practically applied
in a clinical context - the analysis is complex, and it is not clear how this would
be incorporated into a routine clinical examination."

A routine clinical examination can easily be extended to include proposed objective
analysis of the morphological status. We remind that only the raw image of meibomian
glands is sufficient to perform the described analysis. In the case of such an
extended examination, the clinician would have access to both: a real picture of the
glands, and to the corresponding morphometric maps. Of course, we do not expect the
clinician to perform this analysis himself. Our method should be included into the
imaging software. Most conveniently to imaging software of commercial meibograph
(sort of module/plug-in). We hope we provide all the necessary technical details of
the technique to make it easier to replicate it by software engineers.

We agree that the analysis is not trivial, but from the clinician's point of view,
what is important is the way of interpreting the morphometric maps produced by our
method. We have significantly expanded the "Discussion" section with additional
information on how to read those maps so that the reader has a better understanding
of their interpretation. 

Another clinically useful feature of presented method is that morphometric maps can
also be used to define and automatically estimate the values of new meibo-scores. We
added new Fig.8 to show the potential definition of such score. Of course, the
usefulness of any specific parameter determined this way in the clinical practice
must be demonstrated by the future dedicated and systematic tests.

"4. This is important to justify, noting the conclusion (line 316): "To conclude, our
method for isolating individual morphometric components from the original Meibomian
image can be of invaluable aid in the diagnostic process. It may help clinicians to
see in which part of the eyelid disturbance is taking place and also to quantify it
with a numerical value providing essentially insight into disease pathophysiology."
The results presented do not really support the 'invaluable aid' comment and should
be modified as such."

The new section “Conclusion” was introduced where we summarize our results in a
broader way. The statement on 'invaluable aid' was softened.

"5. Can the authors comment on associations between the eyelid meibography 'heatmaps'
generated using the local analysis, and symptoms reported by the patients. Thank
you."

The connection between the observed gland distortion and the patient’s symptoms was
already recognized and described in the literature. We have cited appropriate papers
in our manuscript [9-11, 14-15, 24-25, 32-34]. This observation was based on manual
measurements of different morphological features. The aim of our work was not to
search for a new correlation between gland’s morphological change and particular
symptom, but to provide a tool for automatic and objective characterization of local
gland distortion. This tool allows for quick and automatic mapping of defined
morphological parameters in an objective manner, not requiring manual measurements.
We hope that in the near future the systematic research using this tool may speed up
and objectify the measurement of morphological features providing deeper
understanding of the connection between gland distortion and various clinical
measures. 

We addressed this issue in several places in the significantly expanded "Discussion"
section and in the "Conclusions" section.

"6. A careful edit for English grammar and expression in the entire manuscript would
improve the communication; there are sections that are not well expressed."

We have reviewed the manuscript to improve language usage. In addition to minor
corrections, many new sentences appeared and old sentences have been rewritten to
improve their readability without changing their meaning.

Response to the points raised by Reviewer #1:

"1. Please include details of the original ethics approval, from which the images
were obtained.

It would also be helpful to understand the general inclusion/exclusion criteria of
the original study population"

New subsection named “Subjects” in “Materials and methods” was introduced where
details on ethics approval, inclusion/exclusion criteria and number of examined
patients were included.

"2. Materials and Methods – the meibomian gland images were graded 1-3 on two
consecutive

days and the results were well correlated. However, it would be helpful if the
authors could clarify how many researchers were involved in the grading, and how the
images were classified if the grades assigned were different on the two days"

Only one experienced optometrist was involved in the grading of meibomian images and
the rating was done on two consecutive days. If the grades assigned were different
on the two days, the images were reanalysed again to make a proper decision based on
the presented criteria. 

We have added the appropriate sentences in the section “Meibographic images” to
better clarify this point.

"3. Table 1 – please add confidence intervals or other statistical analyses to this
table. Otherwise,

it is difficult to determine whether these proportions are significantly different
from one another"

95% Confidence intervals were added to the table. Details on their evaluation was
also introduced into the Supplementary Materials document (S7 Appendix).

"4. Line 211 belongs in the Materials and Methods section, but further detail needs
to be added

regarding the “home-built meibographic imaging equipment” used. How does this compare
to commercially available imaging tools"

Standard IR retro-reflection meibograph is very simple device. Actually it is
ordinary camera adapted to work with infrared light. The quality of the acquired
images is determined just by the quality of the lens, resolution and dynamical range
of the camera. The parameters of our system (resolution 20pix/mm and 8-bit dynamics)
were sufficient to record a clear image of the basic structure of the Meibomian
glands, that is, the feature we were interested during our study. We added new
figure (Fig.1) which shows an exemplary raw meibographic image acquired by our
system. As one can see it does not look much different from what is usually
presented as obtained with commercial instruments. We did not test our method on
images acquired with such devices. It is certain that the quality of the images
provided by these instruments will not be worse than ours. Therefore, we are
confident that the described method will work also for that case. Surely, with
better equipment better results can be expected.

More information on imaging technique used for meibography was introduced into
“Meibographic images” section of “Materials and methods” as well as in dedicated “S1
Appendix. Meibomian gland images acquisition” supplementary document.

"5. Line 121 – the sentence beginning “In next section” is unnecessary"

The sentence was removed

"6. The manuscript should remove all personal pronouns e.g. line 121 “we”, line 132
“our”"

Personal pronouns were removed

"7. Line 316 – please soften the conclusion “invaluable aid” – this is too strong and
not necessarily supported by the data"

The new section “Conclusion” was introduced where we summarize our results in a
broader way. The statement on 'invaluable aid' was softened.

"8. Do the findings correlate with any ocular symptoms or other signs of dry
eye?"

The aim of our work was not to search for a correlation between gland’s morphological
change and particular symptom, but to provide a tool for automatic and objective
characterization of local gland distortion. Of course, the motivation for creating
such a tool is a strong evidence that the meibomian gland morphological changes
correlate with the ocular symptoms and signs of the dry eye. We have cited
appropriate papers in our manuscript [9-11, 14-15, 24-25, 32-34]. The presented
method allows for quick and automatic quantification of defined morphological
parameters in an objective manner, not requiring manual measurements. While we do
not provide any evidence for this, this logical sequence implies that the reported
results should correlate with ocular symptoms. 

We hope that in the near future the systematic research using this tool will provide
such a proof and will confirm the clinical usefulness of the proposed approach. 

"

9. What is the clinical utility of this analysis system? Is it something that can be
easily

incorporated into clinical practice? What do the findings actually mean, since no
associations are discussed relating the health of the glands and other signs and
symptoms of dry eye?"

Distortion of the glands is a well-known indicator of the health of the glands and it
is considered to be a symptom of early stage dry eye disease. Our finding concerns
the quantitative and automatic assessment of the degree of deformation of the
glands. Additionally, morphological parameters are presented in the form of spatial
color-coded maps.

A routine clinical examination can easily be extended to include the proposed
objective analysis of the morphological status. We remind that in principle only the
raw image of meibomian glands is sufficient to perform the described analysis. Of
course, we do not expect the clinician to perform this analysis himself. Our method
should be included into the imaging software. Most conveniently to imaging software
of commercial meibograph (sort of module/plug-in). In the case of such an extended
examination, the clinician would have access to both: a real picture of the glands,
and to the corresponding morphometric maps. We have significantly expanded the
"Discussion" section with additional information on how to read those maps so that
the reader has a better understanding of their interpretation.

Another clinically useful feature of presented method is that morphometric maps can
also be used to define and automatically estimate the values of new meibo-scores. We
added new Fig.8 to show the potential definition of such score. Of course, the
usefulness of any specific parameter determined this way in the clinical practice
must be demonstrated by the future dedicated and systematic tests.

We have expanded the "Discussion" section with many additional descriptive sentences,
hoping that the potential clinical utility will be better understood.

Response to the points raised by Reviewer #2:

"1. How was MG function scored?"

The aim of our research was to develop a method for automatic and objective
determination of local changes in the morphology of Meibomian glands. For such a
task, scoring of the MG functions was not necessary. Therefore, the function of the
MG and other ocular symptoms and signs of the dry eye disease were not scored. To
make it clear, we've added a sentence to the "Meibographic images" section. 

Of course, it would be very useful to determine the correlation between morphological
changes and other features that determine the condition of the glands. This task is
beyond the scope of our work. However, there is a strong evidence that the meibomian
gland morphological changes correlate with the ocular symptoms and signs of the dry
eye. We have cited appropriate papers in our manuscript [9-11, 14-15, 24-25, 32-34].
Therefore, we can hope that the parameters estimated by us may also be useful in
diagnosis of the ocular symptoms and the image analysis method will make the
procedure faster and more objective.

"2. How user-friendly this method is?"

Our meibography system is based on the usual retroreflective approach. It has been
mounted on a Topcon SL-D701 slit lamp allowing the recording of meibographic images
of the upper and lower eyelids during a routine eye examination. Image analysis
method described in the article can be easily integrated into an image acquisition
software. What is important for the clinical practice this method is fast and is not
only user-friendly but also patient-friendly causing no discomfort during the
procedure and providing the images which can be easily demonstrated to the patients.
Information on this point was added to the conclusion section.

"3. Adding 1 or 2 sample images to the manuscript will help readers better understand
and visualize to what extent authors are referring to deformity of glands."

In the "Materials and methods" section, a new image has been added (Fig. 1), which
shows the effect of image preprocessing and marks the areas considered by the
optometrist as distorted.

"4. Also, I would like to know about the complete tarsal area what was exposed to
calculate MGs height. Height of MGs may differ with eversion specially for lower
lids as we have less surface area."

In our work, the measure of distortion was strictly defined by derivatives of the
orientation and the width of the glands and did not require knowledge of their
length. Therefore, the height of the glands was not calculated. However, even if our
approach focuses on the shape of the glands (rather than their dimensions) it was
very important to expose the eyelid properly, especially in order to correctly image
the distal areas of the eyelid. We used only the images of the best quality and
presenting the largest area of the tarsal plate containing the Meibomian glands.

Although we have always collected a set of meibographic images of the upper and lower
eyelids for each patient, we only used images of the upper eyelid to test our
method. This decision was made only because of the larger surface of the upper
eyelid on which the shape changes can be easier to visualized. In the further study
the lower eyelids should be examined as well. 

"5. Good but do comment on usefullness of this method. What is the clinical utility
of this technique?"

As mentioned before, distortion of the glands is a well-known indicator of the health
of the glands and it is considered to be a symptom of early stage dry eye disease.
Therefore the detailed morphometric analysis is needed to establish what kind of
changes are crucial in the proper diagnostic procedure. Our finding concerns the
quantitative and automatic assessment of the degree of deformation of the glands.
Additionally, morphological parameters are presented in the form of spatial
color-coded maps. Compared to the other methods our algorithm allows to indicate
subtle changes in the MG morphology, which is important for the proper follow up
examinations.

A routine clinical examination can easily be extended to include proposed objective
analysis of the morphological status. Our method should be included into the imaging
software. Most conveniently to imaging software of commercial meibograph (sort of
module/plug-in). Then, the clinician would have access to both: a real picture of
the glands, and to the corresponding morphometric maps indicating where in the
eyelid there is a specific morphological change (change in angle or change in
width). We have added new sentences into the "Discussion" section on how to read
those maps so that the reader has a better understanding of their
interpretation.

Another clinically useful feature of presented method is that morphometric maps can
also be used to define and automatically estimate the values of new meibo-scores. We
added new Fig.8 to show the potential definition of such score. Of course, the
usefulness of any specific parameter determined this way in the clinical practice
must be demonstrated by the future dedicated and systematic tests.

We have significantly expanded the "Discussion" section and introduced additional
information into “Conclusion” section, hoping that the potential clinical utility
will be better understood.

"6. Tortuosity is related more to the twisting or curved MGs. In literature, there
are several methods to measure tortuosity and deformity. I would like to know more
about the method authors have used and its reliability as well. Better to describe
with sample images."

So far, all the recognized in the literature methods to measure tortuosity and
deformity of the meibomian gland consider only individual the most affected glands
as a representation of the entire eyelid area. The simplest approach is to simply
count the number of distortion events. In the new Fig.1 we have marked, for example,
the regions considered by the optometrist to be distorted. Here we have limited
ourselves to only two features: change of gland direction and glands narrowing. With
our method, the values of these features (or their derivatives) can be estimated
automatically for each position on the eyelid. This not only makes the analysis
procedure more objective and faster, but also makes it possible to define new
meibo-scores. An example of two such scores that quantify the percentage of the
eyelid area covered by deformed regions of the glands is shown in the new Fig.
8.

"7. There are various ways to quantify Meibomian gland images particularly
three-dimensional Fourier domain OCT. How this 2D Fourier transform is better?"

The method of 3D Fourier domain OCT is just another tool for collecting images of the
gland structure. It is imaging technique, not an analytical technique. Here, the
Fourier transformation is used to speed-up acquisition process. This imaging
technique is much more sophisticated than simple retroflection of usual meibographs
(after all it is OCT method). The data are better quality (although the field of
view is rather small) and provide information also on the depth of the gland
(instead of just its surface studied by usual retroreflection meibographs). So
surely, 3D OCT tool provide a lot of high-quality data. But again, it does not
provide ways for quantitative analysis. In order to extract information about any
gland feature (e.g. drop-out volume or gland orientation), the data from 3D OCT
should also be subjected to appropriate analysis. This is the area of application of
our method. Our work does not involve new method for acquiring images (we used usual
retroreflection method), but rather a new way of their analysis focused on automatic
and objective quantification of glands distortion. This is where Fourier transform
came in handy. We are confident that our method can be successfully used also to
analyze data from 3D OCT instruments.

to Reviewers.docx
---

## [Decision Letter · Decision Letter 1]

13 Jun 2022

2D Short-Time Fourier Transform for local morphological analysis of meibomian gland
images

PONE-D-22-01430R1

Dear Dr. Pochylski,

We’re pleased to inform you that your manuscript has been judged scientifically
suitable for publication and will be formally accepted for publication once it meets
all outstanding technical requirements.

Kind regards,

Michele Madigan

Academic Editor

PLOS ONE

Additional Editor Comments (optional):

Reviewers' comments:

Reviewer's Responses to Questions

**Comments to the Author**

1. If the authors have adequately addressed your comments raised in a previous round
of review and you feel that this manuscript is now acceptable for publication, you
may indicate that here to bypass the “Comments to the Author” section, enter your
conflict of interest statement in the “Confidential to Editor” section, and submit
your "Accept" recommendation.

Reviewer #2: All comments have been addressed

2. Is the manuscript technically sound, and do the data
support the conclusions?

Reviewer #2: Yes

3. Has the statistical analysis been performed
appropriately and rigorously? 

Reviewer #2: Yes

4. Have the authors made all data underlying the
findings in their manuscript fully available?

Reviewer #2: Yes

5. Is the manuscript presented in an intelligible
fashion and written in standard English?

Reviewer #2: Yes

6. Review Comments to the Author

Reviewer #2: Authors have addressed all comments. This is a good research area and
authors have efficiently managed to highlight important elements.

7. PLOS authors have the option to publish the peer
review history of their article (what does this mean?). If published, this will
include your full peer review and any attached files.

If you choose “no”, your identity will remain anonymous but your review may still be
made public.

**Do you want your identity to be public for this peer review?** For
information about this choice, including consent withdrawal, please see our
Privacy Policy.

Reviewer #2: No

---

## [Editor Report · Acceptance letter]

15 Jun 2022

PONE-D-22-01430R1 

2D Short-Time Fourier Transform for local morphological analysis of meibomian gland
images 

Dear Dr. Pochylski:

I'm pleased to inform you that your manuscript has been deemed suitable for
publication in PLOS ONE. Congratulations! Your manuscript is now with our production
department. 

Kind regards, 

on behalf of

Dr. Michele Madigan 

Academic Editor

PLOS ONE